# Genome-Wide Profile of Mutations Induced by Carbon Ion Beam Irradiation of Dehulled Rice Seeds

**DOI:** 10.3390/ijms25105195

**Published:** 2024-05-10

**Authors:** Ying Ling, Yuming Zhang, Ming Huang, Tao Guo, Guili Yang

**Affiliations:** 1National Engineering Research Center of Plant Space Breeding, South China Agricultural University, Guangzhou 510642, China; 20213137039@stu.scau.edu.cn (Y.L.); 202113110128@stu.scau.edu.cn (Y.Z.); mhuang@scau.edu.cn (M.H.); 2Heyuan Branch, Guangdong Laboratory for Lingnan Modern Agriculture, Heyuan 517000, China

**Keywords:** *Oryza sativa* L., carbon ion beam (CIB) irradiation, resequencing, single base substitution (SBS), InDels

## Abstract

As a physical mutagen, carbon ion beam (CIB) irradiation can induce high-frequency mutation, which is user-friendly and environment-friendly in plant breeding. In this study, we resequenced eight mutant lines which were screened out from the progeny of the CIB-irradiated dehulled rice seeds. Among these mutants, CIB induced 135,535 variations, which include single base substitutions (SBSs), and small insertion and deletion (InDels). SBSs are the most abundant mutation, and account for 88% of all variations. Single base conversion is the main type of SBS, and the average ratio of transition and transversion is 1.29, and more than half of the InDels are short-segmented mutation (1–2 bp). A total of 69.2% of the SBSs and InDels induced by CIBs occurred in intergenic regions on the genome. Surprisingly, the average mutation frequency in our study is 9.8 × 10^−5^/bp and much higher than that of the previous studies, which may result from the relatively high irradiation dosage and the dehulling of seeds for irradiation. By analyzing the mutation of every 1 Mb in the genome of each mutant strain, we found some unusual high-frequency (HF) mutation regions, where SBSs and InDels colocalized. This study revealed the mutation mechanism of dehulled rice seeds by CIB irradiation on the genome level, which will enrich our understanding of the mutation mechanism of CIB radiation and improve mutagenesis efficiency.

## 1. Introduction

Creating phenotypic variation through spontaneous or artificially induced mutations and excavating mutant genes have been a research hotspot in plant breeding for decades [1]. The mutation frequency of the genome under natural conditions is usually very low, for example, the average spontaneous mutation frequency for each regeneration is only 7 × 10^−9^ in *Arabidopsis thaliana* [2]. The mutation frequency can be increased by tens to hundreds of times by means of artificial mutagenesis treatment, such as carbon ion beam (CIB) irradiation. As one kind of effective high-LET radiation, CIB irradiation has been widely used in mutation breeding for several plant species. CIB irradiation displays an effective induction of DNA double-strand breaks (DSBs), resulting in a broad spectrum of phenotypic variations [3,4,5], which endows it with great value in plant germplasm creation.

Rice (*Oryza sativa* L.) is a staple food for nearly half of the world’s population, and it is also a model crop that has been widely used in physical mutation [6]. Many studies have evaluated the mutagenic effect of CIB radiation on rice from different aspects. CIB irradiation has been used to improve important agronomic traits, such as biotic and abiotic resistance, and some valuable rice mutants have been used in breeding and gene function study, including extremely late heading mutant [7], semi-dwarf mutant [8], low-cadmium rice [6], ultraviolet-resistant mutant [9], salt-tolerant mutant [10], and high temperature-resistant mutant [11]. In order to obtain the expected mutagenic effect, it is necessary to understand the molecular properties of CIB-induced mutation. Previous studies showed that the mutations induced by CIBs are mainly SBSs and small InDels [4,5,12,13], and the single base InDels are more common than the large fragment InDels (≥2-bp) [4]. Some studies also reported that CIBs induced MNVs (multi-nucleotide variants) [1] and SVs (structure variations) [14,15,16], large deletion, and genome rearrangement, such as inversion and translocation [16]. In previous studies, the mutation mechanism of dehulled rice seeds has been rarely reported.

Whole-genome sequencing (WGS) is an efficient approach for variant identification in a mutant at single-nucleotide resolution [17,18], and it provides researchers with a robust tool for characterizing the spectrum of mutations on a whole-genome scale and identifying the causal mutations more efficiently. WGS successfully promoted the identification of induced causal mutations in many plants [1,19,20] and made gene identification much easier. And it also has been used to study mutagenic effects of different mutagens in rice [1,5,21,22] and *Arabidopsis thaliana* [23].

The study of the mechanism of heavy carbon ion beam mutation at the rice genome level will contribute to germplasm creation by radiation mutation. In the present study, we resequenced eight mutagenized rice mutants screened out from the progeny of the CIB-irradiated dehulled rice seeds, and analyzed the genome-wide mutation profiles of all the mutants. This study revealed the mutation mechanism of dehulled rice seeds by CIB irradiation on the genome level, which will enrich our understanding of the mutation mechanism of CIB radiation and improve mutagenesis efficiency.

## 2. Results

### 2.1. Summarized Information of CIB-Induced Mutations

CIB mutagenesis was carried out on H492, H499, and H574, three pure lines of *Oryza sativa* L. ssp. *Indica*. A total of 1000 dehulled and CIB-treated seeds of H492, H499, and H574, respectively, were planted to obtain M_1_ plants. Then, M_2_ seeds from M_1_ plants were collected. Phenotypic screening for visible mutant candidates in the M_2_ or M_3_ population was conducted throughout the whole growth period, and finally, eight lines of M_6_, including three mutants (H494, H495, H496) originated from H492, one mutant (H512) originated from H499, and four mutants (H579, H580, H592, H593) originated from H574, were used for sequencing in follow-up study. The wild-type H492, H499, and H574, respectively, were resequenced as well.

In general, all the mutants displayed visible and heritable traits. The plant architecture and grain type of the lines are shown in Figure 1 and Figure 2. There were significant differences in plant height, grain length, grain width and grain length–width ratio between mutants and wild-type, but the phenotypic mutagenic effects varied among different lines. Compared with the corresponding wild-type, the mutant lines H494, H495, and H496 displayed a significant increase in plant height, whereas the mutant line H512, H579, H580, H592, and H593 showed a decrease in plant height (Figure 1 and Appendix A). The grain size of the mutant lines from H492 significantly increased, whereas the mutant lines from H574 displayed the opposite (Figure 2 and Appendix A).

To investigate the effect of CIB irradiation on the rice genome, we sequenced the mutant lines and the corresponding nonirradiated parental lines. The summarized resequencing information of all the samples is listed in Table 1. On average, 15.8 Gb clean data were obtained for each line and mapped onto the Nipponbare reference genome, resulting in an average sequencing depth of 42.3-fold, and the percentage of genomic coverage with 10× sequencing depth mapped to the reference genome of each line reached 86.6% (Table 1).

Based on the classification of mutation types, reliable mutation data of eight mutants were sorted out, as shown in Table 2. A total of 135,539 variations were detected in these mutant lines treated by CIB irradiation, including 119,579 SBSs and 15,956 InDels, respectively. The number of all SBS mutations is 7.5 times that of InDels, indicating that SBS is the main mutation type (Table 2).

### 2.2. Frequency and Distribution of Mutations for CIB Irradiation

#### 2.2.1. Distribution of Mutations on Chromosomes

To investigate the effect of CIB irradiation on the rice genome, we mix the mutations detected in every single mutant originated from the different WTs. For the sake of description, we name the mutants originated from H492 as Mutation Pond 1 (MP1), the mutant from H512 as Mutation Pond 2 (MP2), and the mutants originated from H574 as Mutation Pond 3 (MP3). Unusual high-frequency (HF) mutation regions were discovered by analyzing the mutation frequency per Mb along the genome of each Mutation Pond. The mutation in the 9~17 Mb region of Chr.12 of MP1 accounts for 37.8% of all mutations, and the mutation frequency in this region is 3.3 × 10^−3^, which is much higher than the average mutation frequency of Chr.12 of 6.3 × 10^−5^, and the same anomalies are also observed in Chr.2 and Chr.6 (Figure 3, Appendix A). The high-frequency region of MP2 is concentrated in the 16~22 Mb region of Chr.3, and the number of mutations in this region accounts for 62.9% of all mutations. The mutation frequency in this region is 2.2 × 10^−3^, which is much higher than the average mutation frequency of Chr.3 of 6.3 × 10^−5^ (Figure 4, Appendix A). The mutation in the 4~9 Mb region of Chr.7 of MP3 accounts for 36.4% of all mutations, and the mutation frequency in this region is 3.3 × 10^−3^, which is much higher than the average mutation frequency of Chr.7 of 4.1 × 10^−5^, and high-mutation-frequency regions are also observed in Chr.3 and Chr.10 (Figure 5, Appendix A).

To investigate the distribution of SBSs and InDels on chromosomes of each single mutant, the number of SBS sites and InDels per 100 Kb was also calculated for each single mutant. Distribution of HF mutation regions varied among a single mutant. For MP1, many more HF regions emerged in H496 than in the other two mutants (Appendix A); for MP2, HF regions of SBSs and InDels occur on Chr. 1 and Chr.3 (Appendix A); for MP3, many more HF regions emerged in H580 than in the other mutants (Appendix A). No same HF mutation region was synpositionally identified on chromosomes among the mutants. But interestingly, the HF regions of SBSs and InDels were almost colocalized.

#### 2.2.2. Theoretical Mutation Frequency of the M_1_ Generation

Although many homozygous mutations have been detected in the genome of M_6_ plants, only a quarter of the mutations in self-pollinated plants can be permanently preserved in their offspring. Therefore, we can roughly estimate the mutation rate of M_1_ according to the probability of heritable mutations in different generations and Mendel’s genetic law referred to in Yang et al. [1]. Assuming that all mutations appear as heterozygotes in the M_1_ generation, and assuming that N is the number of generations, Nn is the number of pure genes and mutations in the Mn generation.
$$N_{1}=N_{n}\frac{2^{n}}{2^{n-1}-1}$$

We estimated the number of M_1_ mutations, and the average mutation rate estimated in this study was 9.8 × 10^−5^. The mutation frequency of different strains is very different. MP1 has the highest mutation frequency of 12.3 × 10^−5^, while MP3 has only 6.0 × 10^−5^ (Table 2).

### 2.3. Characteristics of Mutations Induced by CIB Irradiation

SBSs were revealed as the most abundant mutation. Two types of transitions (Ti) (mutations between the same type of bases, namely purine > purine or pyrimidine > pyrimidine) and transversions (Tv) (mutations between different types of bases, namely purine > pyrimidine or pyrimidine > purine) were identified. In general, CIB induced more transition than transversion in each mutant line, and the average Ti accounts for about 65% of the total substitution. The Ti/Tv ratios of MP1, MP2, and MP3 are 1.2, 1.4, and 1.2, respectively. In general, the proportions of Ti and Tv were similar among the mutants (Figure 6). CIB can induce two kinds of transitions, namely G: A transition and C: T transition. Among all the mutant lines, G > A and C > T were the main transitions, accounting for 66.7% of the total transitions. Eight kinds of transversion, including A > C/T, G > C/T, C > A/G, and T > A/G, were induced. The proportions of the different kinds of transversions in MP1, MP2, and MP3 were almost the same (Figure 7).

The fragment lengths of InDels ranged from 1 to 50 bp. For insertions, the 1 bp insertion and 2 bp insertion were the common types, accounting for 66.5% on average. The same scenario was revealed for 1 bp deletion and 2 bp deletion, accounting for 65.0% on average (Figure 7). The results indicate that the heavy carbon ion beam is more inclined to induce short fragment InDels.

We also analyzed the locations of all the SBSs and InDels on the genome-wide scale. On average, 68.3% of the SBSs were located in the intergenic region of the whole genome, 7.5% in the upstream region, 5.7% in the downstream region, 5.4% in the exonic region, 9.4% in the intron region, 2.7% in the UTR’3, and 1.1% in the UTR’5; 61.2% of the InDels occur in the intergenic region of the whole genome, 9.5% in the upstream region, 7.4% in the downstream region, 2.5% in the exonic region, 13% in the intron region, 4% in the UTR’3, and 2.4% in the UTR’5. Overall, the location distribution patterns of the two types of mutations are similar in each mutant (Figure 8).

### 2.4. Effect of CIB-Induced Mutation on Gene Function

Since the mutations occurring in the exon region are most likely to cause gene dysfunction, we predict the potential mutation effect on gene function by SnpEff, which is a *japonica* rice transcriptome database for gene function prediction. Four types of mutation effects were categorized, including synonymous mutation, nonsynonymous mutation, and nonsense and frameshift mutation. Among the above four types, nonsynonymous mutations only bring in single amino acid change, which may impair protein function. In the gene coding region, an increase or decrease by one or a few base pairs (not multiples of 3) will lead to frame shift mutation, which leads to great changes in protein sequence and even produces non-functional proteins truncated by terminators. Nonsense mutation refers to the mutation of a codon encoding an amino acid into a stop codon by a change in one base, resulting in early termination of peptide chain synthesis. A total of 149 frame shift mutations and 75 nonsense mutations were identified, with an average of 28 such high-impact mutations per line (Table 3).

We specifically analyzed genes with gene function affecting mutations including nonsynonymous mutation and nonsense mutation in each mutant, since these mutations more likely lead to gene dysfunction or even phenotypic variations. A total of 21 mutated genes were found in six mutants, including H494, H495, H496, H512, H580, and H592 (Table 4). These genes are functionally involved in regulating architecture, grain shape, or panicle shape. Seven of these genes were located in HF mutation regions. Four of these genes have still not been cloned. Many more functionally affected genes were discovered in H496 and H592. These results provide some clues to the identification of mutated genes in each mutant.

### 2.5. Verification of the Identified Mutation Sites

To verify the accuracy of resequencing data, six mutation sites were randomly selected for Sanger sequencing. The sequencing results confirmed the 7 bp deletion occurred in exon 3 and a single base substitution occurred in intron 3 of *Os03g0576600* in H512 (Appendix A), 1 bp deletion occurred in intron 6 of *Os08g0425500* in H512 (Appendix A), 2 bp deletion occurred in exon 1 of *Os11g0112050* in H580 (Appendix A), and the insertion of 2 bp in exon 1 of *Os10g0174548* (Appendix A) and the insertion of 7 bp in exon 1 of *Os10g0163290* (Appendix A) in H580 were also confirmed by Sanger sequencing, which is consistent with resequencing data.

## 3. Discussion

The genome-wide profile of mutations induced by CIB irradiation in dehulled rice seeds was studied by genome-wide resequencing technology in this study. CIB irradiation induced SBSs and InDels, and SBSs were the most abundant type of mutation, similar to studies of CIB-induced mutations in other plants, such as *Arabidopsis* and *Brachypodium* [5,13]. However, differences were also revealed in some aspects due to different pretreatment in the present study.

### 3.1. The Mutation Rate of Dehulled Rice Seeds Induced by Carbon Ion Beam

According to the number of mutations revealed by the whole-genome sequencing of rice plants, we estimate that the carbon ion (^12^C^6+^ ions at 100 Gy (LET = 80 keV/µm) irradiation of dehulled dry seeds leads to an average theoretical mutation frequency of 9.8 × 10^−5^/bp, which is surprisingly higher than that in previous studies. Dry rice seeds were irradiated with 40 Gy carbon ion (LET:76 keV/μm), and the estimated mutation frequency was 2.7 × 10^−7^ per base [8]. On the other hand, the whole-genome sequencing analysis of the M_6_ line of “Hitomebore” rice seeds irradiated with 30 Gy carbon ion (LET: 107 keV/μm) estimated that the mutation rate in M_2_ was 2.4 × 10^−7^ per base [14]. The prominently higher mutation rate in our study may mainly be the result of two causes. One is the dosage variation. The dosage used in this study is comparatively higher. Different doses of carbon ion radiation lead to different methylation levels and different biological effects [24,25]. The mutation rate increases with the increase in dose below 150 Gy, and the mutation effect decreases with the increase in dose above 150 Gy [16]. The mutation rate induced by 80 Gyd carbon ion beam is higher than that induced by 40 Gy carbon ion beam [26]. The relatively higher radiation dosage used in this study may lead to higher mutation frequency. Another cause is the premutagenesis dehulling of rice seeds, which may directly expose the embryo to irradiation. “Naked” seeds are more susceptible to mutagenesis, which may further increase the mutation rate in our study. To date, dehulled rice seeds have seldom been used for irradiation studies. Our study may give a clue that dehulling of rice seeds may enhance the mutation frequency. More experiments can be conducted to verify the effect of this mutagenesis seed treatment.

### 3.2. SBSs and InDels Are the Two Main Mutation Types Occurring under CIB Irradiation

SBSs and InDels are two types of mutation identified in all the mutants, and the number of SBSs is 7.5 times that of InDels, indicating that SBSs are the most abundant type of mutation. Previous studies also reported that C ion irradiation tends to induce large fragments of InDels [27] or MNVs (multiple nucleotide variants) [25]. A WGS analysis of 11 mutants in *Arabidopsis* derived from CIB irradiation indicated that single base InDels were more prevalent than larger InDels (≥2 bp) [5]. Our result also showed that single base transition is the main form for SBSs; the short fragment InDels (1-bp and 2-b) are the main form for InDel.

Many abnormal regions of high-frequency mutation regions were also discovered in the present study. This uneven distribution of mutations induced by CIB irradiation was also observed in the previous research [1]. Theoretically, every single base on the genome possesses the same mutability under irradiation, but actually, mutations are not completely randomly occurring. The mutation frequencies of some parts are obviously higher than the average; these are called mutation hotspots [28]. Studies have shown that the point mutation rate of mutation hotspots is at least 20 times higher than that of other regions of the same DNA molecule [29]. The mutation rate of high-frequency mutation regions is even 100 times higher than the average frequency in our study. The reason for the formation of mutation hot spots is still not fully understood. “Common fragile sites” (CFSs) are believed to be present on the genome, and are particularly prone to breakage and instability during mitosis [30]. CFSs might be densely distributed in the mutation hotspots, which lead to the occurrence of high-frequency mutations in these regions on the genome. But un-synpositional HF mutation regions on chromosomes among different mutants in this study might indicate the randomness of CFS occurrence. In these high-frequency mutation regions, we also found that the SBSs and InDels were almost colocalized, which means that the high-frequency regions of SBSs are same as those of InDels.

### 3.3. Gene Function-Affecting Mutations Provide Some Clues to Gene Identification

Gene function-affecting mutations, including nonsynonymous mutation and nonsense mutation, are more likely to lead to gene dysfunction or even phenotypic variations. Targeted analysis of these mutated genes may contribute to identification of genes causing trait variation. Genes functionally involved in regulating architecture, grain shape, or panicle shape were discovered in our study, and they can be candidate genes regulating mutated traits. Of course, if whole-genome resequencing analysis can be combined with Bulked Segregation analysis or MutMap, it will surely improve the accuracy of mutated gene identification.

## 4. Materials and Methods

### 4.1. Plant Material, Mutagenesis, and Mutant Screening

CIB mutagenesis was carried out on H492, H499, and H574, three pure lines of *Oryza:sativa* L. ssp. Indica. The water content of the dry seeds was 15%, which was measured by seed moisture meter. The dehulled seeds of the above three lines were exposed to ^12^C^6+^ ions at 100 Gy (LET = 80 keV/µm) generated by the Heavy Ion Research Facility in Lanzhou (HIRFL) at the Institute of Modern Physics, Chinese Academy of Sciences (IMP-CAS). In the paddy field of South China Agricultural University, irradiated seeds and corresponding wild-types were planted. M_2_ seeds from individual M_1_ plants were collected. We screened visible mutant candidates in the M_2_ or M_3_ population throughout the whole growth period according to the screening criteria ($|\left(Mutated\;trait\;value-Wild\;tpye\;trait\;value\right)÷\;Sdev\left(Wild\;tpye\;trait\;value\right)|\geq 3$), and the candidate mutants were planted in a separate line and bagged for selfing to obtain at least 500 seeds in the subsequent M_3_ to M_6_ generation. Finally, eight M_6_ lines induced by CIB irradiation were used for sequencing.

### 4.2. Whole-Genome Sequencing and Clean Read Filtering

Genomic DNA was extracted from a single plant of each mutant line. The rice genomic DNA was extracted using the cetyl-trimethylammonium bromide (CTAB) method and quantified using a NanoDrop ND-1000 spectrophotometer (Thermo Scientific, Wilmington, NC, USA). Total genomic DNA was extracted from samples and at least 3 µg genomic DNA was used to construct paired-end libraries with an insert size of 300–400 bp using Paired End DNA Sample Prep kit (Illumina Inc., San Diego, CA, USA). These libraries were sequenced using the novaseq6000 (Illumina Inc, San Diego, CA, USA) NGS platform at Genedenovo company (Guangzhou, China). Quality trimming is an essential step to generate high confidence of variant calling. Raw reads would be processed to obtain high-quality clean reads according to four stringent filtering standards: removing reads with ≥ 10% unidentified nucleotides (N); removing reads with > 50% bases having phred (Phred score, Qphred) quality; removing reads aligned to the barcode adapter.

### 4.3. SNP and InDel Identification

To identify SNPs and InDels, the Burrows–Wheeler Aligner (BWA) was used to align the clean reads from each sample against the reference genome with the settings ‘mem 4 -k 32 -M’, where −k is the minimum seed length, and −M is an option used to mark shorter split alignment hits as secondary alignments [31]. Variant calling was performed for all samples using the GATK’s Unified Genotyper. SNPs and InDels were filtered using GATK’s Variant Filtration with proper standards (-Window 4, -filter “QD < 2.0 || FS > 60.0 || MQ < 40.0”, -G filter “GQ < 20”) and those exhibiting segregation distortion or sequencing errors were discarded. In order to determine the physical positions of each SNP, the software tool ANNOVAR 2020 [32] was used to align and annotate SNPs or InDels. The reference genome used in the study is Ensembl_release47_IRGSP-1.0_Nipponbar (http://plants.ensembl.org/Oryza_sativa/Info/Index?db=core, accessed on 28 November 2022.).

#### Variant Annotation and Gene Function Prediction

To identify genes affected by each induced mutation, we used the SnpEff [33] and Rice 7 databases (https://sourceforge.net/projects/snpeff/files/databases/v4_2/, accessed on 3 March 2023). Theoretical mutation frequency of the M_1_ generation was estimated according to the method used in Yang et al. [1].

## Figures and Tables

**Figure 1 ijms-25-05195-f001:**
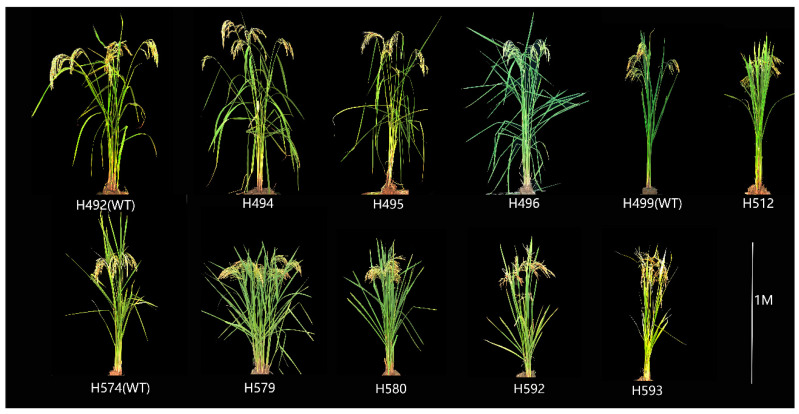
Comparison of plant architectures between wild-type and mutants.

**Figure 2 ijms-25-05195-f002:**
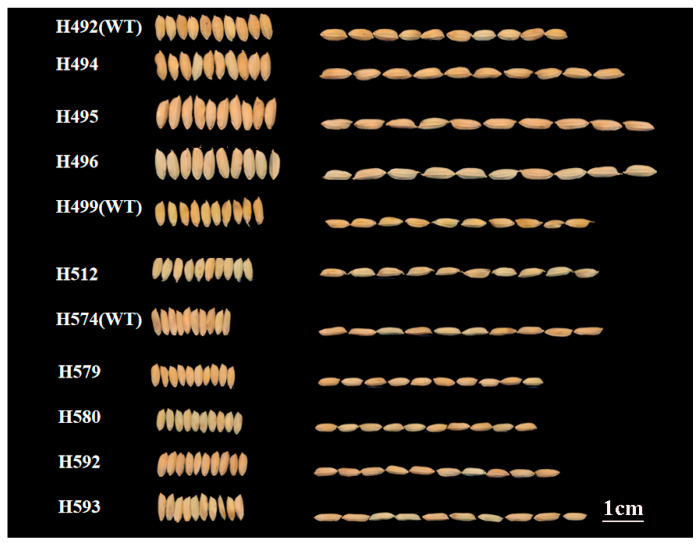
Comparison of grain shape between wild-type and mutants.

**Figure 3 ijms-25-05195-f003:**
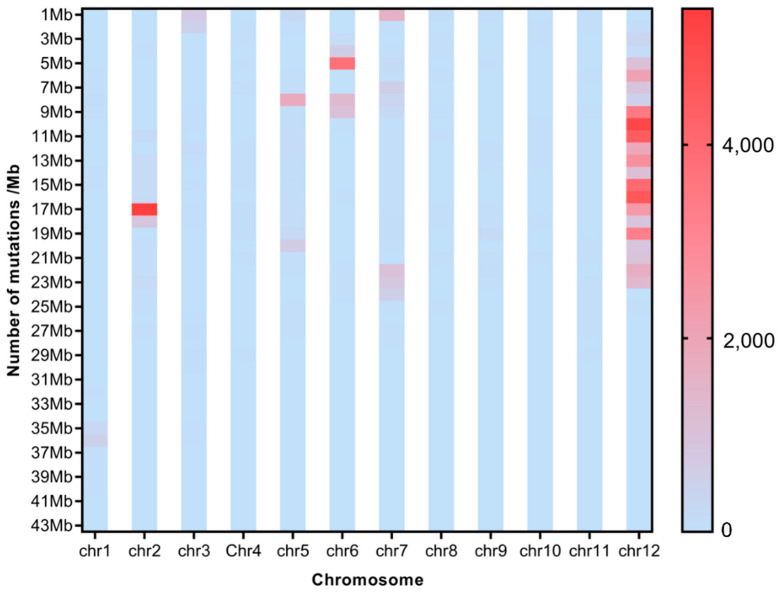
Average mutation distribution of the mutants induced from H492.

**Figure 4 ijms-25-05195-f004:**
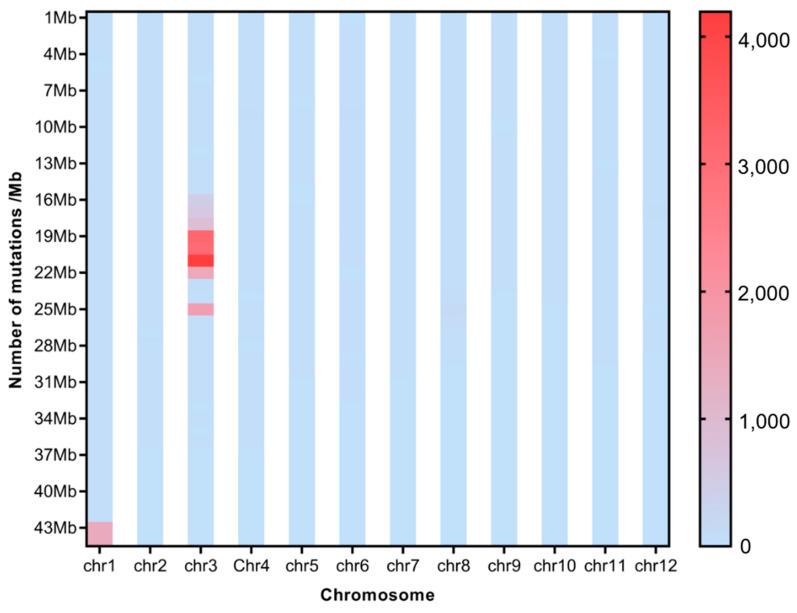
Average mutation distribution of the mutants induced from H499.

**Figure 5 ijms-25-05195-f005:**
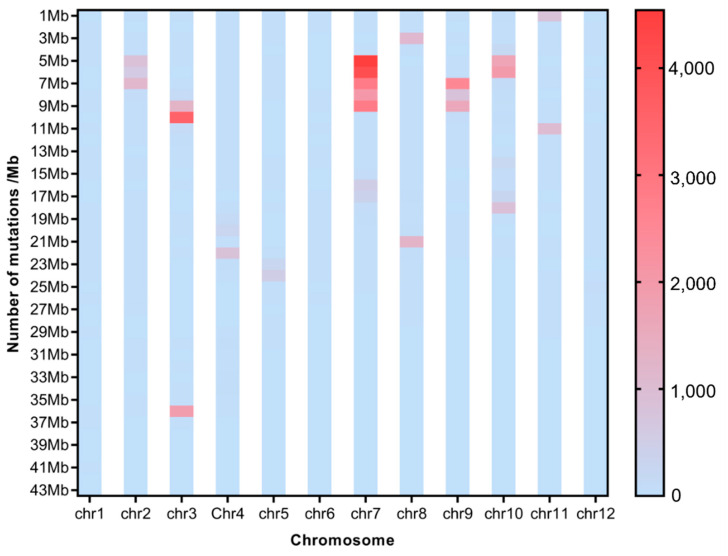
Average mutation distribution of the mutants induced from H574.

**Figure 6 ijms-25-05195-f006:**
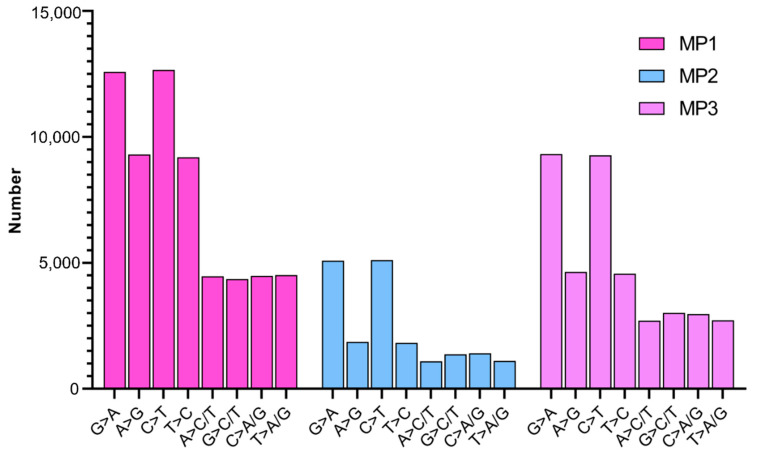
Nucleotide preference in single-nucleotide mutations.

**Figure 7 ijms-25-05195-f007:**
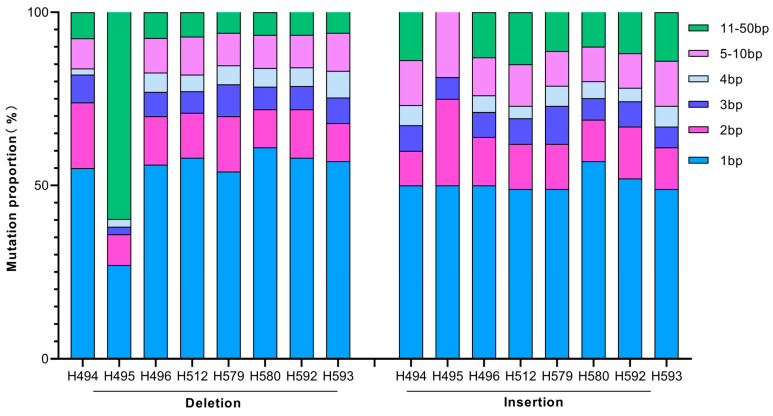
Number of Insertion-Deletion fragments of different length.

**Figure 8 ijms-25-05195-f008:**
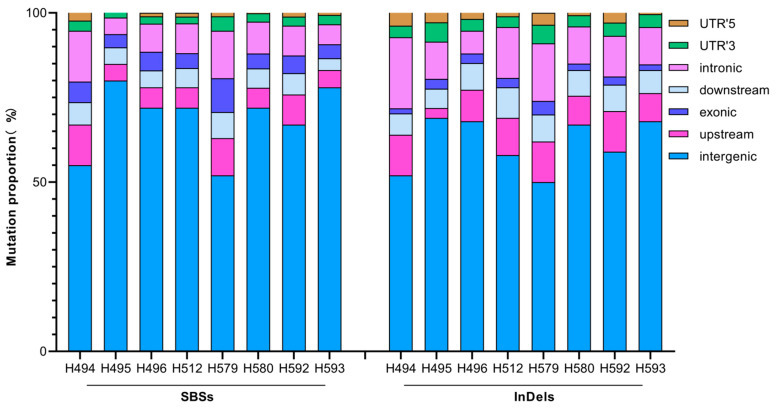
Location distribution of SBSs and InDels induced by CIB irradiation.

**Table 1 ijms-25-05195-t001:** Comparison between wild-type and mutant sequencing data and genome.

Type	Line	Sequenced Generation	HQ_Clean_ Data(bp)	Theoretical Genome Coverage	Genome Coverage of 10× Sequencing Depth (%)	Genome Coverage of 20× Sequencing Depth (%)	Genome Coverage of 30× Sequencing Depth (%)
**WT**	H492	M6	16.0	42.9	93.79	90.22	80.01
**Mutant**	H494	M6	16.2	43.5	88.05	77.08	62.57
**Mutant**	H495	M6	16.2	43.3	88.72	77.47	62.03
**Mutant**	H496	M6	16.4	44.1	89.67	79.73	65.36
**WT**	H499	M6	13.6	36.5	79.55	64.22	45.77
**Mutant**	H512	M6	14.2	38.0	80.63	66.28	48.47
**WT**	H574	M6	15.6	41.9	85.99	79.55	64.71
**Mutant**	H579	M6	15.9	42.7	86.62	79.67	64.11
**Mutant**	H580	M6	16.9	45.3	86.95	80.7	67.93
**Mutant**	H592	M6	16.2	43.4	86.38	79.72	65.78
**Mutant**	H593	M6	16.2	43.4	86.51	80.5	66.75
**Average**			15.8	42.3	86.6	77.7	63.0

**Table 2 ijms-25-05195-t002:** Summary of mutation information of individual lines.

WT	Mutant	Total	SBSs	InDels	The Estimated Mutations in M_1_	Mutation Rate in M_1_
H492	H494	2455	2101	354	4910	12.3 × 10^−5^
	H495	145	110	35	290	
	H496	66,813	59,315	7498	133,626	
H499	H512	21,101	18,860	2241	42,202	11.3 × 10^−5^
H574	H579	8503	7257	1246	17,006	6.0 × 10^−5^
	H580	6196	5528	668	12,392	
	H592	23,689	20,548	3141	47,378	
	H593	6633	5860	773	13,266	
Average		16,942	14,947	1995	33,884	9.8 × 10^−5^

**Table 3 ijms-25-05195-t003:** Different types of mutation effects on gene function of each mutant.

Category	Mutant	Synonymous Mutation	Non-Synonymous Mutation	Nonsense Mutation	Frameshift Mutation
MP1	H494	63	62	1	3
	H495	0	2	1	0
	H496	1411	1728	39	75
MP2	H512	328	465	10	22
MP3	H579	327	302	11	15
	H580	105	130	3	7
	H592	593	496	7	20
	H593	123	116	3	7
Total		2950	3301	75	149
Average		368.8	412.6	9.4	18.6

**Table 4 ijms-25-05195-t004:** Candidate genes with function-affecting mutations.

Mutant	Gene ID	Whether in HF Mutation Region	Gene Description	Possibly Affected Phenotype	Cloned or Not
H494	Os06g0247500	No	Pyrophosphate-fructose 6-phosphate 1-phosphotransferase	Plant height, grain shape	Yes
H495	Os02g0278400	No	Cytochrome P450	Plant height, grain shape	No
H496	Os05g0374200	No	Cytokinin oxidase/dehydrogenase 9	Plant height, grain shape, setting percentage	Yes
H496	Os01g0826400	No	WRKY transcription factor	Plant height, grain shape	Yes
H496	Os07g0214300	No	α-amylase/trypsin inhibitor	Grain shape	Yes
H496	Os07g0235800	No	APETALA2-like transcription factor	Grain shape	Yes
H496	Os07g0583700	No	WRKY transcription factor 78	Plant height, grain shape	Yes
H496	Os12g0428600	Yes	HECT-domain E3 ubiquitin ligase	Plant height, grain shape	Yes
H496	Os12g0496900	No	GRF-INTERACTING FACTOR 2	Plant height, grain shape	No
H496	Os12g0552600	No	Grain Weight (qTGW12a)	Grain shape	Yes
H512	Os01g0972800	No	WRKY transcription factor	Plant height, grain length	No
H512	Os03g0430000	Yes	defective embryo sac1	Setting percentage	Yes
H512	Os05g0547850	No	Programmed cell death 5	Plant height, grain shape	Yes
H580	Os04g0396500	No	lax panicle2	Panicle type	Yes
H592	Os03g0267800	No	Ubiquitin-interacting motif-containing ubiquitin receptor	Grain shape	Yes
H592	Os03g0274800	Yes	Receptor-like cytoplasmic kinase	Plant archetechture, setting percentage	Yes
H592	Os07g0192300	Yes	NARROW LEAF 11	Plant archetechture, panicle type	Yes
H592	Os07g0232100	Yes	B3 domain transcription factor	Panicle type	Yes
H592	Os07g0233300	Yes	Alfin-like gene	Grain length	Yes
H592	Os07g0235800	Yes	APETALA2-like transcription factor	Grain shape	Yes
H592	Os09g0281900	No	Mediator subunit gene	Plant height, grain shape, setting percentage	No

## Data Availability

All data generated or analyzed during this study are included in this published article/Appendix A. The WGS data reported in this study have been deposited in the Genome Sequence Archive (Genomics, Proteomics & Bioinformatics) in the BIG (Beijing Institute of Genomics, Chinese Academy of Sciences) Data Center and are publicly accessible at http://bigd.big.ac.cn/gsa, (accessed on 18 October 2023), reference number PRJCA019968.

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
