# Peer review of "Genome-Wide Profile of Mutations Induced by Carbon Ion Beam Irradiation of Dehulled Rice Seeds"

_ijms, 2024, doi:10.3390/ijms25105195_

Round 1

Reviewer 1 Report

Comments and Suggestions for Authors

 The objective of the current study conducted by Ling et al., entitled “Genome wide Profile of Mutations Induced by Carbon ion Beam Irradiation of Dehulled Rice Seeds” was to investigate Carbon ion beam (CIB) irradiation, as a physical mutagen environmentally friendly for mutation breeding. In this investigation,  authors conducted a resequencing of 8 mutant lines derived from dehulled rice seeds irradiated with CIB. Within these mutants, CIB exposure led to the detection by sequencing of 135,535 variations. The research is intriguing,  but it requires further elaboration, especially in the discussion chapter.

 The authors wrote: “Many abnormal regions of high frequency mutation regions were also discovered in the present study”.  The authors should delve deeper into this point, especially considering that different lines exhibit distinct hot spot positions mapping on different chromosomes.

Minor issues:

- The legends of figures and tables should be extended to better explain to the reader.

e.g. What is CK?

-  English should be improved

e.g. Page 13       “ . Irritated seeds and the corresponding wild type …”

Comments on the Quality of English Language

Moderate revision

Author Response

Thank you very much for taking the time to review this manuscript. Thank you for giving us valuable advice. We have revised and resubmitted the manuscript. In the current version, we have compared the high frequency mutation region among the different mutants. No same HF mutation region was synpositionally identified on chromosomes among the mutants in this study. Specific discussion on this point has been added in the discussion part of the revised manuscript.

Reviewer 2 Report

Comments and Suggestions for Authors

This manuscript described investigation of mutation sites and its distribution trends at genome sequence level for eight rice mutant lines derived from carbon-ion irradiation treatment. The authors clearly revealed a total of 135,353 mutation sites in the whole genomes in the eight mutant lines, and estimated that a part of the mutation sites have effects of changing phenotypes in rice plants, because there were a lot of nonsynonymous mutations and nonsense mutations in the genome sequences of the eight mutant lines. As the authors mentioned, it is important to elucidate molecular mechanisms of mutations by carbon-ion irradiation for developing novel rice cultivars in further breeding. However, the number of mutant lines used in this study is too small, and the authors did not describe significant novelty for relationships between mutated genes and phenotype alteration in the mutant lines. These additional genetics information would be necessary to demonstrate the significance and effectiveness in this study.  Therefore, I recommend major revision of the current version of this manuscript.

Author Response

Thank you very much for taking the time to review this manuscript. Thank you for giving us valuable advice. We have revised and resubmitted the manuscript. In the current version, we have specifically analyzed genes with gene function affecting mutations including nonsynonymous mutation and nonsense mutation in each mutant since these mutations more likely lead to gene dysfunction or even phenotypic variations. Many genes functionally involved in regulating architecture, grain shape or panicle shape were identified in different mutants. These results provide some clues to the identification of mutated genes in each mutant. We have also added some discussions on this point in the the revised manuscript.

Round 2

Reviewer 2 Report

Comments and Suggestions for Authors

This manuscript was clearly revised by the authors based on the previous comments. I did not send some of my comments last time, so I'll resubmit them. However, the current version of this manuscript is well revised, so no response is required this time. Please use them as a reference for your future research.

1) This study collected polymorphism sites in eight mutant lines derived from three indica rice accessions. To investigate trends for distribution of the mutation sites in the rice genome, the eight lines were too small. The authors should additionally encode genome sequences in more numbers of mutant lines in this study.

2) The authors should indicate full names of MNV and SV at the Introduction section.

3) The authors should indicate explanations for origins of each mutant lines in the 1st line at the Results section. You could describe similar sentences here that are with 1st line at the Materials and Methods section.

4) The authors used the genome sequence of Nipponbare as reference genome for collecting mutation site. I think that the other reference genome of indica rice cultivar is good for your study, because there are large differences for genome sequence variations between Japonica and indica rice cultivars.

5) Please add 'H' in the legend of Figure S2.

Author Response

Thank you so very much for your constructive suggestions on our future research. We have revised the manuscript according to your suggestions.